# UV Laser-Induced Phototransformations of Matrix-Isolated 5-Chloro-3-nitro-2-hydroxyacetophenone

**DOI:** 10.3390/ijms24021546

**Published:** 2023-01-12

**Authors:** Magdalena Pagacz-Kostrzewa, Karolina Mucha, Maria Wierzejewska, Aleksander Filarowski

**Affiliations:** Faculty of Chemistry, University of Wrocław, F. Joliot-Curie 14, 50-383 Wrocław, Poland

**Keywords:** photoisomerization, argon matrices, nitro group, DFT, FT-IR

## Abstract

Conformational changes of 5-chloro-3-nitro-2-hydroxyacetophenone were studied by experimental and theoretical methods. Phototransformations of the compound were induced in low-temperature argon matrices by using UV radiation, which was followed by FT-IR measurements. Two types of changes within the molecule were detected: rotations of the hydroxyl and acetyl groups. A new conformer without an intramolecular hydrogen bond was generated upon irradiation with λ = 330 nm, whereas the reverse reaction was observed at 415 nm.

## 1. Introduction

The matrix isolation method is a technique used for experimental studies which was suggested for the first time by Pimentel, Whittle, and Dows [1,2,3]. This method has been successfully developed for decades and it has been described in a number of reviews [4,5,6,7,8,9,10,11]. It is noteworthy that the inter- and intramolecular interactions under matrix isolation conditions are often specific and differ from interactions in gas phase, solutions, or in solid state [12]. These peculiarities have been precisely described in a review by Barnes [13]. The matrix isolation method can also be effectively used in studies of photo and thermo processes, since the structure of metastable states can remain fixed [14,15,16,17,18]. The application of this method is advantageous for interpreting vibrational spectra thanks to the absence of rotational and phononic modes observed in gas phase and solid state, respectively [19,20]. 

Notably, the studied phenomenon of intermolecular proton transfer plays an important role in chemical and biological processes [21]. The studied compounds with intramolecular proton transfer have been applied as active media for both photochemical lasers and photo-stabilizers [22], sensors of metal ions [23], and also for storage of information and optics switchers [24]. Moreover, they can act as fluorescent sensors to measure transmembrane potential in cell membranes [25]. In addition, the nitroaromatic hydroxyketones have been widely employed for manufacturing dyes, pesticides, herbicides, fungicides, paints, and explosives in the chemical industry [26,27].

In this paper, we are concerned with experimental and theoretical studies of conformational states of the nitro derivative of ortho-hydroxy acetophenone observed under the influence of irradiation. The studied compound can exist in a few metastable states which result from rotations of the nitro and acetyl groups. In our previous paper, we revealed a possible competition between OH⋯ONO and OH⋯O=C hydrogen bonds in solutions [28]. Moreover, other studies have shown that the studied compound underwent a phase transition and could exist in two polymorphic forms in the solid state [29]. The novelty of the presented studies is based on revealing and modeling the processes that occur in a molecule, which allow one to efficiently reduce the radiation energy in the UV spectral range. To be more exact, the studied acetophenone derivatives are used as protectors (UV filters) against solar radiation [30]. Therefore, supporting the importance of studies on the influence of radiation on the isomerization process in acetophenones that can show the way to quench solar radiation energy.

## 2. Results and Discussion

### 2.1. Energy Minima and Transition States of CNK on the S_0_ Potential Energy Surface (PES)

The obtained experimental data proved that the studied compound exists in several conformational states under argon matrix isolation. In order to define the structures of the conformers, DFT calculations (B3LYP/6-311++G(3df,3pd)) were performed that revealed the presence of four stable conformers (**A**–**D**) of the studied compound. The transition states joining the energy minima on the S_0_ PES were optimized at the same level of theory allowing us to estimate energy barriers between the conformers. The ZPE-corrected potential energy diagram for the interconversions of CNK is shown in Figure 1. Conformer **A** with an OH⋯ONO intramolecular hydrogen bond is the most stable one, meanwhile conformer **B** with an OH⋯O=C intramolecular hydrogen bond has only slightly higher energy (energy difference of ΔE_AB_ = 4.55 kJ mol^−1^). Conformer **C,** which is also OH⋯ONO bonded and has the turned acetyl group, is less stable than conformer **A** by 14.30 kJ mol^−1^. It is notable that all three conformers mentioned above contain an intramolecular hydrogen bond, which strengthens their stability as compared with conformer **D**, characterized by the highest energy and the absence of a hydrogen bond. The energy difference between conformers **A** and **D** is large and equals ΔE_AD_ = 50.65 kJ mol^−1^. The obtained values suggest that the two most stable conformers, i.e., **A** and **B,** are likely to be detected in the gas phase as well as in low-temperature matrices after deposition. It can also be assumed that under the conditions of low-temperature matrix isolation, without the influence of external radiation, conformers **C** and **D** can hardly be observed.

### 2.2. Matrix Isolation Spectrum of CNK and Its Comparison to the Theoretical Predictions

The analysis of the obtained infrared IR spectra in the mid-infrared range for CNK definitely confirms that two conformers, i.e., **A** and **B,** are observed under the argon matrix conditions directly after the deposition at 15 K (10 K for measurements). This conclusion is based on the observation of pairs of bands in the spectrum of the studied compound. Figure 2 (black trace, upper panel) shows the selected ranges of the IR spectrum obtained after the deposition of CNK/Ar matrix at 15 K/10 K. According to our previous studies [28,29], the bands at 1700.0 cm^−1^ and at 1668.0 cm^−1^ are assigned to stretching vibrations of the carbonyl group (ν(C=O)), not involved and involved in the intramolecular hydrogen bond, respectively. In turn, bands observed at 1539.0 and 1550.5 cm^−1^ are due to the asymmetric stretching vibration of the nitro group (ν_as_(NO_2_)). These bands are assigned to conformers **A** and **B** in accordance with the presented results of the DFT calculations. As mentioned above, these conformers are close in energy (ΔE_AB_ = 4.55 kJ mol^−1^, Figure 1) and both comprise hydrogen bonds between the hydroxyl and nitro groups, and the hydroxyl and acetyl groups, respectively.

### 2.3. Phototransformations of CNK upon UV Laser Irradiation

Here, we present the results of our studies on the influence of UV irradiations on conformational states of CNK under matrix isolation conditions. A broad range of wavelengths was used in order to find some promising photochemical reactions. The most significant changes were observed at λ = 330 nm irradiation. Figure 2 shows selected parts of infrared spectra of CNK isolated in an argon matrix: after deposition and upon UV irradiations. As shown, upon irradiation at 330 nm, the intensity of the ν(C=O) band at 1700.0 cm^−1^ increases and the one at 1668.0 cm^−1^ decreases. This result highlights the UV-induced conformational change between conformers **A** and **B** that leads to an increase in the **A** species at the expense of the **B** species. This conclusion is supported by the observation of similar changes in the spectral range of the asymmetric stretching vibrations of the nitro group (ν_as_(NO_2_)). Indeed, intensity of the band at 1550.5 cm^−1^ (conformer **B**) decreases, whereas that of the band at 1539.0 cm^−1^ slightly increases. As presented in Figure 1, the transformation of **B** into **A** requires consecutive rotations of two fragments of the molecule. The first is the rotation of the OH group, which causes formation of conformer **C,** followed by rotation of the acetyl group leading to conformer **A**. If the rotations were reversed, the intermediate product would be conformer **D**.

A careful examination of the ν(C=O) range of the spectrum obtained upon irradiation at 330 nm reveals the appearance of a new band at ca. 1715 cm^−1^ (blue star in panel **b1**, Figure 2). According to the DFT calculations, this band is assigned to the stretching vibration of the carbonyl group (ν(C=O)) which is not involved in a hydrogen bond. This assignment is approved by an analysis of the 3550–3480 cm^−1^ spectral region. In this region, a new band appears at higher wavenumbers at 3508.5 cm^−1^, which is assigned to the ν(OH) stretching vibration of the non-hydrogen bonded hydroxyl group (Figure 2, red trace in panel **a1**). This spectral pattern corresponds well to the theoretically predicted spectrum of conformer **D,** confirming that this structure is formed in the CNK/Ar matrix upon irradiation at 330 nm. It was not observed in the spectra obtained directly after deposition due to a significant difference in energies between form **D** and the most stable form **A** as well as a high energy barrier for their interconversion (ΔE_AD_ = 54.71 kJ mol^−1^ on the S_0_ potential energy surface, see Figure 1). The appearance of conformer **D** becomes possible upon the electronic excitation of conformer **B** at 330 nm.

Interestingly, conformer **C** is not observed in the experimental spectra under the influence of 330 nm radiation despite its predicted energy being much smaller than that for conformer **D**. A possible explanation of this experimental observation can be found by examining the energy diagram shown in Figure 1. It shows that conformer **C**, when formed at 330 nm, is expected to undergo a barrierless transformation and relax easily into the most stable conformer **A**.

When the CNK/Ar matrix, previously irradiated at λ = 330 nm, was subjected to subsequent irradiation at 415 nm, the reverse reaction was observed, manifested by an apparent increase in conformer **B** bands and a slight decrease in conformer **A** bands (see Figure 2, blue trace, middle panel). Simultaneously, the bands of conformer **D** disappeared. The most probable mechanism of the photoisomerization begins with the rotation of the CH_3_-C=O group of conformer **A** around the C-C bond that leads to the formation of conformer **C**, which is not detected under the condition of the current experiment. However, the barrier for the **C**→**B** transformation, involving the rotation of the OH group, is low enough to be overcome by the applied energy, which results in the formation of conformer **B**.

In order to confirm our interpretation of the processes occurring in the system, irradiation at λ = 415 nm was also performed on the freshly deposited CNK/Ar matrix. The expected **A**→**B** conversion was observed in the spectra. However, an insignificant increase in the newly formed conformer **B** indicates low efficiency of the process. This phenomenon can be explained by analyzing the potential energy diagram (Figure 1) that shows two possible transformations for conformer **A**, i.e., the rotation of either the acetyl group or the hydroxyl group, which result in the formation of **C** and **D**, which are immediately converted to form **A**. It is possible because of very low energy barriers of 0.06 and 4.06 kJ mol^−1^ for **C**→**A** and **D**→**A** isomerizations, respectively. A summary of the UV-induced, relatively complex, conformational changes of CNK in argon matrices is presented in Figure 3.

## 3. Materials and Methods

The 5-chloro-3-nitro-2-hydroxyacetophenone (CNK) was purchased from Merck (Darmstadt, Germany) (99%) and crystallized from methanol before use. The high-purity argon gas (5.0) was obtained from Linde (Dublin, Ireland). In order to obtain matrices containing CNK, the crystalline sample was allowed to sublimate at ca. 380 K in a small electric oven located inside the vacuum vessel of the cryostat. The CNK vapors mixed with a large excess of matrix gas (argon) were deposited onto a CsI window kept at 15 K in a closed-cycle helium cryostat (APD-Cryogenics, Macungie, PA, USA). The temperature was measured directly at the sample holder using a silicon diode sensor working with a digital controller (Scientific Instruments, model 9650-1, West Palm Beach, FL, USA). Infrared spectra were recorded, at 10 K in the range 4000–400 cm^−1^ with a resolution of 0.5 cm^−1^, by means of Fourier transform IR spectrometers (Bruker IFS 66 (Bruker Optik, Ettlingen, Germany) or Nicolet iS50 (ThermoFisher Scientific, Madison, WI, USA,)) equipped with an MCT detector cooled with liquid N_2_. Matrices were irradiated with the tunable UV light provided by the frequency-doubled signal beam of a pulsed (7 ns) optical parametric oscillator Vibrant 355 (Opotek Inc., Carlsbad, CA, USA) pumped with a pulsed Nd:YAG laser (Quantel Technologies, Villejust, France). The experiments started using λ = 430 nm light, and then proceeded with a gradual decrease in the output wavelength. After each irradiation, an infrared spectrum of the matrix was taken.

All calculations were performed with the Gaussian 16 program package [31]. The CNK energy minima and transition states were optimized in a vacuum at the B3LYP/6-311++G(3df,3pd) level of theory [32,33,34,35]. The associated force constant matrixes were calculated at the same level of theory to evaluate harmonic frequencies and zero-point vibrational (ZPE) corrections. The obtained structures were visualized using the GaussView software [36]. To account for anharmonicity effect, the calculated wavenumbers were scaled by 0.920 and by 0.971 within 4000–2800 cm^−1^ and 2800–400 cm^−1^ regions, respectively. The scaling factors were obtained by fitting the computed wavenumbers of CNK to the experimental wavenumbers, separately, for two spectral regions.

## 4. Conclusions

The matrix isolation technique coupled with FT-IR spectroscopy was used to identify different conformers of 5-chloro-3-nitro-2-hydroxyacetophenone (CNK) and to follow possible conformational changes of the compound. Among four stable conformers of CNK, the two most stable forms (**A** and **B**) with intramolecular hydrogen bonds were identified in argon matrices after deposition. Using two different wavelengths of the tunable UV laser beam, several isomerization processes were induced in the matrices. For the first time, it was shown that the irradiation of CNK at 330 nm leads to the appearance of a new conformer **D** without an intramolecular hydrogen bond, which is energetically less stable than conformer **A** or conformer **B** by ca. 50 kJ mol^−1^, according to the DFT calculations. The amount of conformer **D** increases at the expense of form **B**. Simultaneously, the **D**→**A** process is observed leading to an increase in the amount of the most stable conformer **A**.

## Figures and Tables

**Figure 1 ijms-24-01546-f001:**
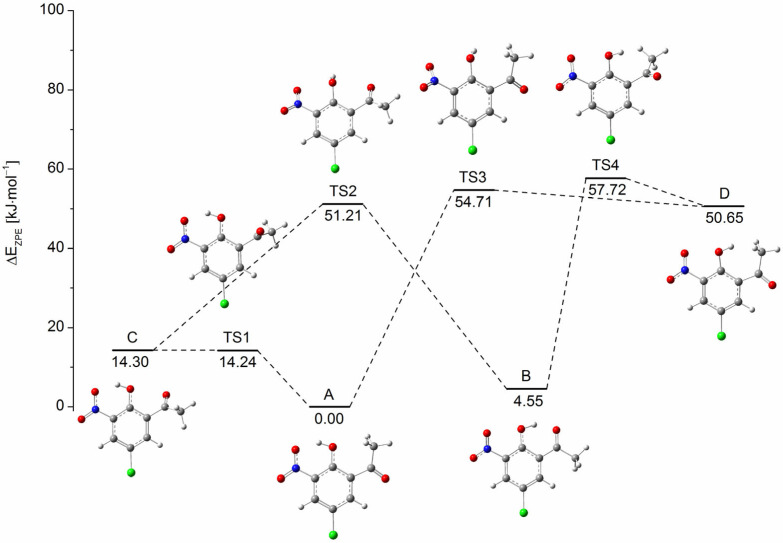
ZPE corrected potential energy diagram for the CNK isomerizations calculated at B3LYP/6-311++G(3df,3pd) level of theory.

**Figure 2 ijms-24-01546-f002:**
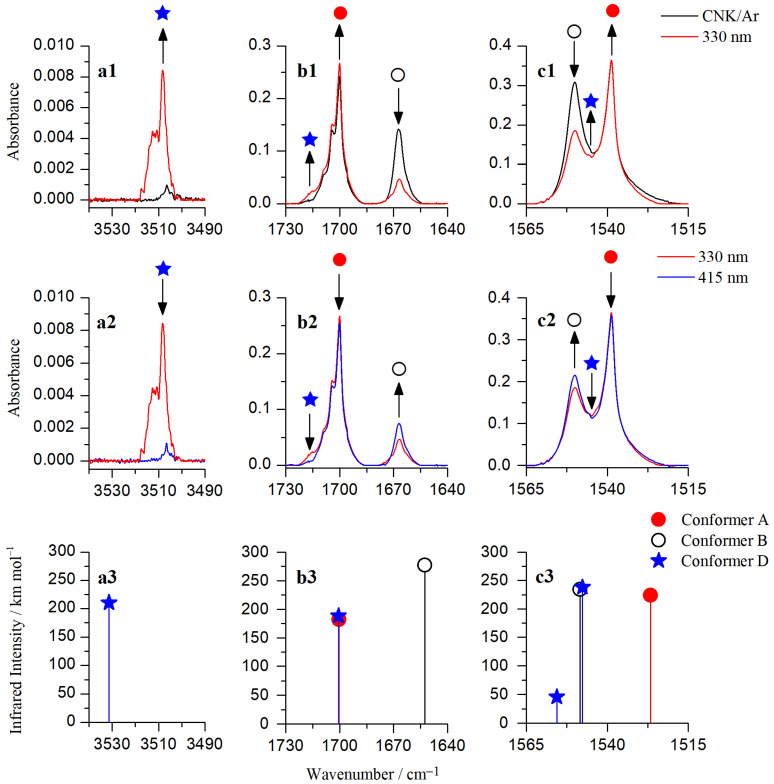
Selected parts of infrared spectra of 5-chloro-3-nitro-2-hydroxyacetophenone isolated in an argon matrix: (**a1**–**c1**) After deposition at 15 K (10 K for measurement) (black spectra); (**a1**–**c1**,**a2**–**c2**) upon irradiation at 330 nm (red spectra) and at 415 nm (blue spectra), respectively; (**a3**–**c3**) stick spectra, calculated for conformers **A**, **B**, and **D** at the B3LYP/6-311++G(3df,3pd) level of theory.

**Figure 3 ijms-24-01546-f003:**
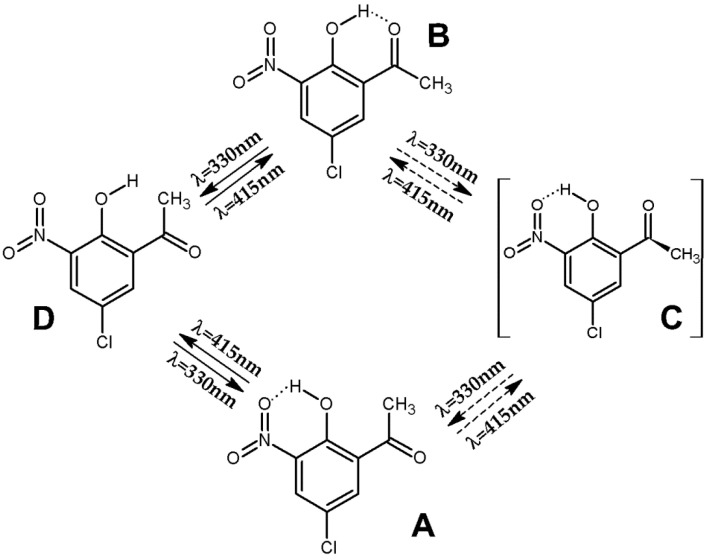
Diagram of photoisomerization processes in 5-chloro-3-nitro-2-hydroxyacetophenone (CNK) upon irradiation at 330 and 415 nm.

## Data Availability

All data associated with this article are included in article.

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
