# Peer review of "UV Laser-Induced Phototransformations of Matrix-Isolated 5-Chloro-3-nitro-2-hydroxyacetophenone"

_ijms, 2023, doi:10.3390/ijms24021546_

Round 1
Reviewer 1 Report
The paper deals with photochemical transformation of 5-chloro-3-nitro-2-hydroxyacetophenone in matrix isolated conditions. Although the experiments and calculations are well done, the overall hypothesis is not sufficiently defended. The authors consider photochemical transformations without considering excited state PES. Without this information most of the conclusions about the possible mechanism are speculative.
I would suggest the following experiments and calculations to be done before to recommend the manuscript for accepting in the IJMS:
1. Excited state PES considering also possibilities for ESIPT;
2. Clear explanation why irradiation at 330 and 415 nm yields different results. This requests considering also the absorption spectra of the isomers;
3. The intro part is very unclear and should be rewritten. Especial importance should be take to explain the novelty in respect of the previous investigations of the same group;
4. The level of self-citing should be reduced.
Author Response
Response to Reviewers
We thank the reviewers for taking the time to evaluate our manuscript. Below are our replies to the issues raised by the Reviewers, and a list of changes and corrections applied to the manuscript. The Reviewers' comments and questions are repeated in italics.
Reviewer 1
- Excited state PES considering also possibilities for ESIPT;
The mechanism of the processes described in the manuscript is our suggestion based on the experimental findings described. We are aware that the excited state PES calculations rightly suggested by the reviewer would show in more detail how the processes proceed. However, such calculations require much more sophisticated methods and as such are beyond the scope of this manuscript. It is also possible that the described isomerizations occur in the vibrationally hot electronic ground state if the molecule is able to access the conical intersection between the lowest singlet excited state and the ground state (M. Pagacz-Kostrzewa, M.A. Kochman, W. Gul, M. Wierzejewska, Phototransformations of 2-aminonicotinic acid resolved with matrix isolation infrared spectroscopy and ab initio calculations. J. Photochem. Photobiol. A-Chemistry, 2021, 410, 113187/1-113187/12.).
- Clear explanation why irradiation at 330 and 415 nm yields different results. This requests considering also the absorption spectra of the isomers;
We performed time-dependent DFT calculations to find out what is the energy of the S0®S1 vertical excitations for the CNK conformers (see table below). Although the results of such calculations are generally burdened with some error, it can be seen that conformer A and conformer B are characterized by clearly different S0®S1 energy. It explains why we looked for the photochemical response of these conformers at different wavelengths.
Table 2. Energy of the vertical S0→S1 excitation DE and corresponding oscillator strength (f) values calculated for the CNK conformers at the B3LYP/6-311++G(3df,3pd) level.
Conformer |
DE (nm) |
f |
A |
389 |
0.074 |
B |
355 |
0.105 |
C |
385 |
0.068 |
D |
342 |
0.070 |
- The intro part is very unclear and should be rewritten. Especial importance should be take to explain the novelty in respect of the previous investigations of the same group;
The first passage of Introduction dwells on the method applied in our studies.
The second passage of Introduction points out why the studied compounds are important in terms of their application.
The third passage of Introduction “predicts” what kind of physical-chemical phenomena could be observed in such type compounds. A fragment
In line with the Reviewers’ recommendations, we completed Introduction with a fragment focusing on the novelty of these studies which lies in the fact that photo-physical phenomena occurring in the molecule will make it possible to develop new materials for industrial application.
The following fragment in the Introduction was added and marked in green color.
“The novelty of the presented studies is based on the revealing and modelling of the processes occurring in the molecule, which allows one to efficiently reduce the energy of radiation in the UV spectral range. To be more exact, the studied acetophenone derivatives are used as protectors (UV filters) against solar radiation [30. Siegel, H.; Eggersdorfer, M. Ketones. Ullmann’s encyclopedia of industrial chemistry. Wiley-VCH, Wienheim, 2002]. Therefore, the studies of the radiation influence on the process of isomerization in acetophenones, which can show the way to quench the solar radiation energy, is very important.”
As for “Especial importance should be take to explain the novelty in respect of the previous investigations of the same group”, the novelty is in the studies of conformers with competing hydrogen bonds exposed to irradiation (our earlier studies involved the investigation of the environment influence on conformational equilibrium in such type compounds, see Refs 28 and 29), and it is dealt with in the manuscript.
- The level of self-citing should be reduced.
We removed five our references and kept two most necessary. One reference was added -[30. Siegel, H.; Eggersdorfer, M. Ketones. Ullmann’s encyclopedia of industrial chemistry. Wiley-VCH, Wienheim, 2002]. The numbering of the references was corrected.
Reviewer 2 Report
In the manuscript are described experimetal and theoretical studies of conformational changes of 5-chloro-3-nitro-2-hydroxyacetophenon.
In the experimental studies the phototransformation of molecular system was induced by UV radiation in low-temperature Ar matrices. Received data of phototransformation were recorded using the FT-IR method.
Conformational studies were also performed at the DFT level using B3LYP functionals and 6-311++G(3df,3pd) basis set. All possible structures were localized on the PES.
Especially interesting and valuable are results of experimental FT-IR studies.
Theoretical calculations were performed at good level of accuracy.
The subject presented in manuscript is important and interesting.
I do not have critical remarks.
Author Response
Dear Reviewer
We kindly thank the Reviewer for evaluating our manuscript.
Best regard,
Aleksander Filarowski
Round 2
Reviewer 1 Report
In the present form the manuscript is not improved and not suitable to be published. Considering photochemical processes only in the ground state is out of sense.